# Modelling and Refining Neuronal Circuits with Guidance Cues: Involvement of Semaphorins

**DOI:** 10.3390/ijms22116111

**Published:** 2021-06-06

**Authors:** Greta Limoni

**Affiliations:** Department of Basic Neuroscience, University of Geneva Medical Center, Rue Michel-Servet 1, 1206 Geneva, Switzerland; greta.limoni@epfl.ch

**Keywords:** amyotrophic lateral sclerosis, axon pruning, dendritogenesis, neurodegeneration, neuromuscular junction, perineuronal net, plexins, retina, semaphorins, synaptic plasticity

## Abstract

The establishment of neuronal circuits requires neurons to develop and maintain appropriate connections with cellular partners in and out the central nervous system. These phenomena include elaboration of dendritic arborization and formation of synaptic contacts, initially made in excess. Subsequently, refinement occurs, and pruning takes places both at axonal and synaptic level, defining a homeostatic balance maintained throughout the lifespan. All these events require genetic regulations which happens cell-autonomously and are strongly influenced by environmental factors. This review aims to discuss the involvement of guidance cues from the Semaphorin family.

## 1. Introduction

The nervous system is assembled by a complex series of timed events, which rely on genetic regulations happening cell-autonomously as well as in response to environmental stimuli [1]. Failure in accomplishing at least one of major developmental steps leads to disorders that, if not lethal, cause lifelong disability [2,3,4]. The proper establishment of neuronal circuits requires neurons to mature by shaping their morphology and making synaptic contacts with a given partner cell. During a first period, both axons and synapses grow supernumerary and are only subsequently refined through a process called pruning. All these events require cell-autonomous activation of genes, but also involve regulation of spatially and temporally expressed cues in the surrounding environment [5]. Guidance receptors, such as Netrins, Ephrins and Semaphorins, have been long involved in this refining process and studies are continuously bringing evidences on the multiple coordinated activity they play to contribute in organizing an orchestrated circuitry.

Semaphorins (SEMAs, Sema in invertebrates) and Plexins (PLXNs, Plex in invertebrates) are a group of ligand-receptor proteins conserved in invertebrates, mammals and viruses [6,7,8] (Figure 1, Box 1). Better known as major players in guiding axons to their target, they have gained increasing attention as cell polarizers, mediating developmental processes of central nervous system (CNS) patterning such as neurogenesis, neuron and glia migration, and brain and spinal cord midline crossing [9]. This versatility is mainly conferred by their finely tuned spatiotemporal regulation, now known to rely on a plethora of microRNAs active on different cells at different developmental stages [10,11,12].

In light of the most recent findings, this review covers the involvement of SEMAs and PLXNs in final aspects of neural development and refining of circuits. Specifically, discussions focus on neuronal maturation and segregation of dendritic arbors to specific targets, and describe involvements in the processes of synaptogenesis and synaptic plasticity in mammals. A chapter is dedicated to axonal pruning and how SEMAs were shown to remove redundant collaterals in the hippocampus and cortical spinal tract. As example of circuit refinement and consolidation in the peripheral nervous system (PNS), this review gathers current knowledge at the neuromuscular junction of *Drosophila melanogaster*. Finally, a prospective chapter focuses on neurodegeneration and, by bringing the example of amyotrophic lateral sclerosis, is willing to question if understanding mechanisms behind SEMAs and PLXNs can prompt a better knowledge of diseases onset and progress.

## 2. Neuronal Maturation

### 2.1. Parvalbumin-Expressing Interneurons

Parvalbumin (PV)-expressing cells constitute the largest population of cortical inhibitory neurons, able to dampen their own and excitatory cells’ activity with a peculiar non-adaptive fast-spiking electrophysiological behavior [13]. It is possibly because of their firing properties that PV+ interneurons, concomitantly to their maturation, are enwrapped by a protective sheet of extracellular matrix, the perineuronal net (PNN) [14,15,16], which supposedly creates a protection from oxidative stress [17]. Accumulating evidence shows that SEMA3A, already expressed by these cells as they establish in the neocortex [18], binds to chondroitin sulfate proteoglycans, one of the main components of PNNs, to shape and probably stabilize the ensheathing matrix [19,20,21,22]. The enrichment of SEMA3A in this extracellular matrix may restrict neurite growth onto cell bodies of PV+ interneurons, which could explain why emergence of PNNs determines the closure of plasticity windows [23]. In the visual cortex, interfering with SEMA3A at the PNNs reopens the plasticity window [24], a phenomenon that can be beneficial for correction of activity-dependent defects [25].

### 2.2. Dendritogenesis

Along with axonal growth, proper establishment of neural circuitry firstly depends on the development of dendritic arborization. Extracellular cues cover an essential function in this process [26] and, interestingly, disturbing SEMAs expression results in aberrant dendritic development [27,28]. In vitro experiments showed that SEMA3A confers polarization to excitatory neurons [29], with dual effect as suppressor and promoter on axons and dendrites, respectively [30]. More recently, studies are questing the intracellular mechanisms that allow such different outcomes. As previously shown in sensory neuron axons [31], different PLXN intracellular domains mediate divergent SEMA signaling, initiating discrete pathways. That is, dendrites challenged with SEMA3A are prompt to elaborate their arborization through the KRK motif of PLXNA4, which interacts with FARP2 and activates the Rac1 GTPase [32]. By contrast, PLXNA4 does not need to engage this intracellular domain to trigger SEMA3A-dependent repulsion in axons [31,32]. How SEMA3A manages to activate distinct intracellular domains of the same PLXNA4 in different cell compartments remains an open question and it is still unknown whether other players in the complex allow for this selection. In the hippocampus, mice lacking *Sema3a* present abnormal apical branching of CA1 pyramidal cells [33]. Follow-up studies further demonstrated that, in these neurons, SEMA3A phosphorylates Cdk5/p35, which targets the Collapsin Response Mediator Receptors (CRMPs) 2 and 4 for cytoskeletal remodeling [33,34,35,36]. Another study described the requirement of PLXNA1, the expression of which is regulated by activity, in SEMA3A-mediated dendritic modeling. Upon binding, the NRP1/PLXA4 complex recruits FARP1, activating Rac1 and promoting actin assembly [37]. In all of these studies, the disruption of one tassel of these pathways is sufficient to constrain dendritic arborization, indicating the necessity of multiple players in the neuron maturation.

In brain regions where dendrites need to achieve subcellular specificity, SEMAs were shown to regulate branching for laminar segregation. The best example is conferred by the retina, where distinct SEMAs are expressed layer-wise and types of neural cells are in there interconnected vertically and horizontally. Dendrites of horizontal cells making ribbon synapses onto rod photoreceptors express PLNXA4 and SEMA6A throughout the first two postnatal weeks. Deletion of either *Plxna4* or *Sema6a* induces overgrowth of their dendrites into the outer nuclear layer, where photoreceptors have their cell bodies, without mistargeting the latter, but instead increasing the number of ribbon synapses with horizontal cell dendrites remaining in the outer plexiform layer (OPL) [38]. Although this mechanism needs to be further elucidated, it hints a role for PLXNA4/SEMA6A in developing OPL stratification. Another study demonstrated that acute disruption of SEMA4A in the pigmented epithelial cell layer (the outermost, as barrier to the choroid) is sufficient to cause degeneration of photoreceptors, which no longer extend their outer segment and dramatically decrease the OPL thickness [39,40]. The inner plexiform layer (IPL) contains synapses between bipolar cells, amacrine cells and ganglion cells (RGCs), axons of which fasciculate to form the optic nerve. It is segregated in sublaminae, where the outers contain OFF synapses of amacrine cells (i.e., in response to light increase) and the inners contain ON contacts (i.e., in response to decreased contrast) [41]. Layer segregation within IPL was shown to rely on PLXNA4/SEMA6A [42,43] and PLXNA1/3 with SEMA5A/B [44] on different cell types. In *Sema5a^−/−^* and *Sema5b^−/−^* mice, RGCs expressing PLXNA1 and PLXNA3 broadly innervate the IPL, with dendrites also reaching the inner and nuclear layers and the OPL [44]. On the other hand, amacrine cells expressing PLXNA4 confine their dendrites in outer layers of the IPL, given the presence of SEMA6A in the innermost part [42]. Finally, dose-dependent expression of SEMA3A in the ganglion cell layer was shown to be involved in RGCs polarization and in their dendritic and axonal growth. In vitro studies showed that inhibition of SEMA3A by miR-30b allows RGCs to assume a bipolar morphology and promotes axonal growth by blocking the Rho intracellular pathway. High dose of SEMA3A, in contrast, induces dendritic branching [45,46,47].

## 3. Synaptic Plasticity

Functional connectivity between neurons is conferred by synaptic contacts, whose plasticity is a long-life process and underlies higher cognitive capacities [3,48]. During the developmental spurt, synapses are made in excess and proper function of the network depends on selective stabilization of strong contacts and pruning of less-active ones [49]. After this period, synapses continuously undergo turnover (i.e., elimination and new formation), but their overall density on a given dendrite remains stable [50]. Although plasticity is mainly due to activity, other receptors act as modulators in the processes of synaptogenesis and synaptic elimination. Among them, many SEMAs and their cognate PLXNs receptors were described to activate downstream Rho and Rap GTPases well known mediate plasticity [51,52] (Figure 2).

### 3.1. Hippocampus

Given its peculiar connectivity structure, which can be maintained intact in culture over long periods, the hippocampus has been largely used as an advantageous model to study synaptic plasticity [53]. In this model, evidence for SEMAs implications in excitatory and inhibitory synapses modeling were studied for over two decades, serving as the biggest piece of data we dispose so far. SEMA4s are widely expressed in the hippocampus and most of them were shown to contribute to synaptic plasticity in this cortical structure. SEMA4D is expressed in pre-synaptic GABAergic terminals [54], from where it interacts with PLXNBs at the dendritic shafts [55,56]. The extracellular domain of SEMA4D facilitates the formation of GABAergic synapses [57,58] by engaging with trans-synaptic PLXNB1 [54,59]. Here, PLXNB1 regulates downstream Rap and Rho GTPases, expanding the size of Gephyrin clusters and stabilizing the newly formed inhibitory synapses [55,60,61]. Interestingly, intrahippocampal infusion of SEMA4D proteins in mice models for epilepsy was shown to be sufficient to dampen seizure events, by recruiting new Gephyrin clusters through PLXNB1 [60]. Requirement of additional PLXNB2 and PLXNB3 along with PLXNB1 at post-synaptic site was also addressed, and possibly contribute in eliminating neighboring excitatory boutons for homeostatic processes [54,56]. Their mechanisms of action, however, remain elusive. Other SEMA4s, such as SEMA4A, SEMA4B and SEMA4F, are involved in excitatory synaptogenesis. Pre-synaptic SEMA4A on excitatory cells engages post-synaptic PLXNB2 which, in turn, is likely to activate *cis*-membrane SEMA4A for establishing intracellular signaling [54]. SEMA4B and SEMA4F were shown to interact directly on the post-synaptic site, recruited by PSD95 itself [61,62], and thus probably contributing to stabilization of excitatory boutons. It is worth of notice that combinations of SEMA4s and PLXNBs are present both at pre- and post-synaptic sites. Although exact mechanisms are not fully clear yet, it is likely that these complexes work both *cis*- and *trans*-synaptically, in autocrine and paracrine manners, to create a feed-forward and backward loop for bouton formation and, subsequently, for their proper stabilization. SEMA3s were also shown to be involved in synaptic plasticity. Specifically, SEMA3F was demonstrated to regulate spine density on apical dendrites of granule cells in the dentate gyrus through NRP2/PLXNA3 [63]. An additional study highlighted that *crmp2*^−/−^ mice present increased spine density in the dentate gyrus due to aberrant pruning, mimicking results previously found in the absence of at least one among *Sema3f*, *Nrp2* or *Plxna3* [64]. This result indicates the requirement of CRMP2 as downstream target of the SEMA3F/NRP2/PLXNA3 signaling complex. More recently, a study presented interesting results on how SEMAs and PLXNs interact across the lifespan for synaptic plasticity related to memory formation. Endothelial SEMA3G is secreted by blood vessels and regulates synaptic transmission of excitatory neurons in CA1, expressing NRP2/PLXNA4. Mice lacking *Sema3g* present deficient behavior selectively in hippocampus-dependent memory formation and less long-term potentiation at electrophysiological level [65]. This reveals a unique interplay between peripheral systems and brain in regulating the need for plasticity elicited by external factors.

### 3.2. Neocortex

In the neocortex, SEMA3B and SEMA3F were shown to have a role in pruning synapses selectively at apical dendrites, depending on the expression of adhesion molecules [66,67,68,69]. Close Homologous L1 (CHL1) and NrCAM associate to NRP2 to form a surface heterocomplex, stabilizing binding to PLXNA4 or PLXNA3, respectively, and enabling affinity to different extracellular SEMAs. Indeed, a subset of spines co-expressing CHL1/NRP2/PLXNA4 prunes when binding to SEMA3B [69]. Mice knocked-out for CHL1 or SEMA3B present a comparable phenotype of increased spine density on apical dendrites both in visual and prefrontal cortices, and these retain an immature morphology (i.e., thin or filopodia-like) throughout adulthood [69]. By contrast, those spines in which NRP2 clusters with NrCAM are pruned by binding with SEMA3F [67,68]. As shown with structural and functional experiments, SEMA3F induces surface clustering of NrCAM/NRP2/PLXNA3, activating Rac1 and RhoA pathways [70]. Specifically, upon binding, PLXNA3 intracellular domain inactivates Rap1, allowing activation of the RhoA-ROCK1/2-Myosin II pathway for actin remodeling. Concomitantly, Tiam1 is recruited, activating the Rac1-PAK-LIMK-Cofilin pathway to provide for spine destabilization and finally cytoskeletal disassembly [70]. Although the regulatory mechanism remains to be fully explained, it is interesting to notice that both SEMA3B and SEMA3F release is activity-dependent [69,70]. Activity, in turn, induces clustering of adhesion molecules, NRP2 and PLXNAs to the surface of less active spines for their turnover.

## 4. Axon Pruning

Axon pruning is the process of removal of supernumerary or misguided axon branches. Two types of pruning occur during development: small-scale axon terminal pruning, in which axon terminal branches innervating a given target area are removed by competition, and stereotyped axon pruning, in which collateral bifurcations to inappropriate targets are eliminated [71]. Consequent to neuronal maturation, this event, occurring all over the nervous system, regulates refinement of projections and is strongly supported by environmental factors. SEMAs are well-known players in axonal pathfinding [72], and not surprisingly also mediate their pruning.

In the hippocampus, SEMA3F is upregulated by interneurons in the infrapyramidal bundle, a transiently long structure below CA3 where cells from the dentate gyrus extend their mossy fiber branches [73]. Presence of NRP2 and PLXNA3 on these latter allows the recruitment of the Rac-specific GAP β2Chn to the membrane, inducing mossy fibers to only retain axons in the main bundle and pruning the infrapyramidal one [73,74]. As shown in mice lacking *Nrp2* or *Sema3f* [73], the *Crmp2*^−/−^ adult hippocampus maintains the infrapyramidal bundle elongated over the CA3 curvature and mossy fibers establish mature synapses in here [64]. This indicates that, as for dendritogenesis and synaptogenesis, SEMA3F may need to phosphorylate CRMP2 for cytoskeletal disassembly and axonal retraction.

From the neocortex, subcortically projecting neurons extend their axons toward multiple targets in the midbrain and spinal cord. These axons, which form the corticospinal tract, are subsequently pruned to retain only relevant connections. An elegant example is given by layer 5 pyramidal cells of the visual and motor cortices. During development, their axons make multiple collaterals in the superior and inferior colliculi and extend toward the spinal cord. However, with the onset of pruning, PLXNA3, PLXNA4 and NRP2 expression becomes restricted to the visual cortex. Only layer 5 axons from here, then, acquire responsiveness to SEMA3F upregulated in the inferior colliculus and the first tract of spinal cord, making them eventually retain only branching to the superior colliculus [75]. Other SEMAs may be involved in pruning motor axons from the midbrain, but this possible mechanism remains to be investigated. A couple of recent studies demonstrated that motor neuron axons are enriched in PLXNA1 and, along with its ligand SEMA6D, promotes retraction to refine functional synapses onto premotor neurons in the spinal cord through the Bax/Bax-caspase-9 pathway [76,77].

## 5. Synaptic Homeostasis at the Neuromuscular Junction

In the PNS, the neuromuscular junction (NMJ) is probably the most studied structure as example of distal cell-cell signaling and synaptic homeostasis [78,79]. Most of the knowledge we have to date were obtained by genetic manipulations on *Drosophila melanogaster*, its NMJ features being conserved in vertebrates. PlexB is expressed by motor neurons (MNs), whose axons are guided toward individual muscular targets by the converged action of Sema2a and Sema2b during larval periods [80,81,82,83]. Although the formation of synaptic contacts (i.e., active zones) seems to exclusively rely on activity [84], some evidences confer the Semaphorin signaling a central role in their refinement and homeostatic maintenance. Action potentials from MNs enhance secretion of Sema2a by muscles, stabilizing functional synaptic contacts [85,86]. More recently, Sema2b was shown to maintain pre-synaptic homeostatic release through downstream activation of Mical in MNs and interactions with the cytoskeleton [87]. Mical signaling was already known to mediate PlexA-Sema1a repulsion by oxidating and destabilizing F-actin [88,89,90]. It is unclear how Mical mediates potentiation of the active zone in MNs [87], but it is possible that its activation starts an intracellular stabilizing pathway through PlexB itself or regulation of other enzymes to reverse its signaling and induce polymerization instead [91]. Finally, in a model for neuropathies characterized by NMJ denervation [92], reducing levels of PlexB or overexpressing Sema2b rescue behavioral and neurobiological failures of GlyRS mutant flies [93]. This could indicate that, in MNs, GlyRS mediates PlxnB expression, which proper functioning regulates Sema2b release, further confirming the requirement of both anterograde and retrograde signaling to keep homeostatic active zones.

## 6. Neurodegenerative Diseases: Is There a Lesson to Learn from Development?

Neurodegeneration is a progressive and non-reversable process affecting the physiology of CNS and/or PNS plasticity [94,95]. These diseases are mainly linked to genetic mutations, which eventually manifest as pathological when affecting new protein synthesis and homeostasis [95,96]. Increasing studies on animal models for neurodegenerative diseases highlighted that dysfunctions in SEMAs and PLXNs dynamics could create a substrate for irreversible damage, by inhibiting regeneration [97] or inducing cell death [98], for example. SEMAs and PLXNs dysregulations are found in several neurodegenerative disease, such as multiple sclerosis [99,100,101,102,103,104,105], spinal muscular atrophy [106], Huntington [107] and Parkinson’s [108,109,110,111] diseases (Table 1). Here the focus is brought to amyotrophic lateral sclerosis (ALS) and the puzzling dilemma of SEMAs-related degeneration.

### Amyotrophic Lateral Sclerosis

ALS is a fatal neurological disease characterized by progressive degeneration and loss of spinal and cortical MNs [112]. Causes of ALS remain poorly understood. Mutations in over 40 different genes, mainly regulating RNA metabolism and autophagy (such as *SOD1*, *FUS*, *OPTN*, *C9orf72*), were identified to contribute to 10% of etiology, while the remaining majority of cases is considered due to sporadic events [112]. Despite this vast heterogeneity and multiple clinical phenotypes [113], analyses point to convergent mechanisms damaging axonal protein homeostasis and inducing cytoplasmatic aggregation, with activation of inflammatory pathways [95,114,115,116].

Studies in mice carrying mutation for *Sod1* (i.e., SOD1-G93A), one of the genes implicated in ALS pathogenesis, convergently showed the etiology at the NMJ, of which denervation causes retrograde and progressive death of spinal MNs, interneurons and ultimately corticospinal cells [117,118,119,120,121]. As reported above, homeostatic maintenance of the *Drosophila* active zones at the NMJ was shown to be supported by Semaphorins and Plexins [87], opening speculations that these guidance receptors may contribute to the disease pathogenesis.

Axon-sequencing on MNs derived from mouse Embryonic Stem Cell (mESC) found downregulation of *Nrp1* transcripts in SOD1-G93A mice [122], while its expression in the spinal cord was increased [123]. Moreover, analyses on injured skeletal muscles showed upregulation of SEMA3A both in the muscle itself and in terminal Schwann cells (TSCs) [124,125]. Given that NRP1 was shown to be a critical protein in attracting and fasciculating corticofugal axons and MNs [126,127], along with the repellent role of SEMA3A [128], it has been long thought that this unbalance at the NMJ would trigger the irreversible degeneration seen in ALS. However, data we have so far are discordant [129,130,131,132], mainly suggesting a protective role of SEMA3A in the PNS instead [125,133,134,135]. The reason behind these opposite results may be due to different experimental approaches and to the unspecificity of the techniques implied. Rescued phenotypes were observed, for example, with injections of anti-NRP1 [136] or muscle overexpression of miR126-5p [131]. However, NRP1 can bind different SEMA3s and function as coreceptor for multiple PLXNs or other adhesion molecule, rendering the statement of cause-effect of difficult interpretation. Furthermore, miRNAs are known to regulate multiple gene expression at once, thus we cannot exclude that overexpression of miR126-5p blocked SEMA3A release along with activation or deactivation of other proteins implied in NMJ homeostatic regulation. Other studies stating the irrelevance of SEMA3A/NRP1 in peripheral axon regeneration [135] or in ALS pathophysiological development [134] used animal models carrying mutations for these genes not restricted to a temporal window and/or a given cell type, possibly masking collateral interactions during the multiple developmental processes and between different cells. Future studies must be directed in understanding whether ALS-related gene mutations alter homeostatic mechanisms relying on SEMA3A. Disruption of SEMAs-dependent interactions among the different cell types at the NMJ could establish a fragile premise for the irreversible degenerative process [115,116,137,138,139].

## 7. Concluding Remarks

SEMAs and PLXNs constitute a class of guidance receptors that, for their dynamic regulation and versatile affinities, modulate steps of nervous system development and confer protection after brain or spinal cord insult [9]. In here, the focus is brought to functions of these molecules in neuronal maturation, from dendritic elaboration to synaptic formation and maintenance. These processes happening toward the end of the developmental period are crucial for setting homeostatic functions that will regulate brain activity along the lifespan. Dysfunctional SEMA-PLXN signaling pathways or mutations in genes carrying these proteins were found in neurodevelopmental disorders, such as autism [146,147,148,149,150], Rett syndrome [151,152] and schizophrenia [153,154] to name a few, and an increasing number of studies indicates that they may mediate irreversible damages in neurodegenerative diseases, having either early or late onset. All these evidences together indicate the important supply SEMAs and PLXNs give during neuronal circuit organization and hint to a protective homeostatic role throughout the whole life. However, the high complexity of their signaling possibilities and outcomes makes their roles still not fully clarified and much remains to be unveiled. Future studies should continue in understanding molecular basis of Semaphorin interactions during neuronal development and homeostatic processes. Contributions in the field may help in deciphering pathophysiology of diseases manifesting upon circuits maturation and may make us closer in understanding yet unknown pathogenesis mechanisms.

Box 1.SEMA-PLXN interactions.SEMAs constitute a family of ligands classified in eight subclasses (1 to 7 and V) (Figure 1) according to their phylogenetic analysis, anchorage to the cellular membrane and C-terminal structure [6]. SEMA1s, 2s, 3E, 4s—7 and V directly interact with PLXN receptors [7], intracellular domain of which recruits effectors that start discrete pathways for cytoskeletal rearrangement [155,156]. SEMA3A-D, 3F and 3G, instead, are targeted by Neuropilin (NRP) receptors 1 and/or 2, which form a heterocomplex with PLXNs [157,158].Structural studies demonstrated SEMAs *trans*-cellular signaling depends on dimers binding PLXNs couples to form bivalent 2:2 complexes [159]. SEMAs are also able to bind monomeric PLXN receptors without, however, enabling intracellular pathway activation [159,160]. An additional signaling mechanism is given by additional *cis*-membrane interaction, when both SEMA ligand and PLXN receptor are present on the same cell surface. Sema1b and SEMA6s monomers can bind Plex/PLXNAs monomers forming a functional heterodimer and modulating *trans*-cellular interactions [161]. Affinity of PLXNs to bind *cis*- over *trans*-SEMAs is likely due to their conformation at resting state, where a monomeric ring-like receptor can bind head-to-head a single adjacent SEMA [160,161].

## Figures and Tables

**Figure 1 ijms-22-06111-f001:**
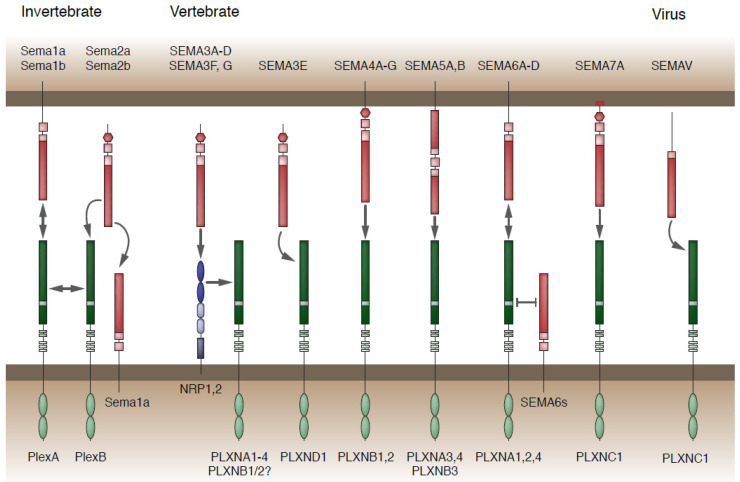
Semaphorins (SEMA/Sema) and Plexins (PLXN/Plex) diversity. Sema2s, SEMA3s and SEMAV are secreted proteins, while the others are membrane anchored (SEMA7A presents an additional glycosylphosphatidylinositol). Sema1s, Sema2s and Sema5c (not pictured here as its function is unknown) are present in invertebrates, while SEMAV is found in viral genome. All the remaining (SEMA3s to SEMA7A) are vertebrate Semaphorins. In the nervous system, Semaphorins are signaling directly through Plexins. Exception is made for SEMA3A, SEMA3B, SEMA3C, SEMA3D, SEMA3F and SEMA3G which are bound by Neuropilins (NRP) 1 or 2, and form a receptor complex with PLXNAs to initiate an intracellular pathway.

**Figure 2 ijms-22-06111-f002:**
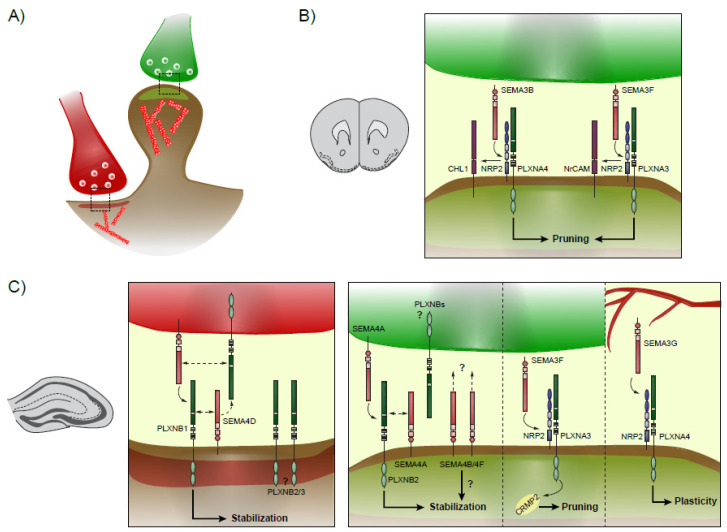
Synaptic plasticity in the neocortex and hippocampus. Schematics showing the different known Semaphorins-Plexins interactions during spinogenesis and plasticity. (**A**) Excitatory axons are colored in green, while inhibitory axons are colored in red, and respectively form excitatory and inhibitory synapses onto the post-synaptic neuron. (**B**) In the neocortex, excitatory synapse pruning is mediated by SEMA3B and SEMA3F, which signals through CHL1/NRP2/PLXNA4 and NrCAM/NRP2/PLXNA4, respectively. (**C**) In the hippocampus, inhibitory synapses are formed and stabilized by a complex interaction between SEMA4D and PLXNBs, present at both pre- and post-synaptic sites. Excitatory synaptogenesis is modulated by SEMA4A and PLXNB2 and may additionally rely on SEMA4B and SEMA4F. On the other hand, pruning is initiated by NRP2/PLXNA3 binding to SEMA3F. Finally, SEMA3G secreted by blood vessels interacts with NRP2/PLXNA4 to confer plasticity for new memory formation.

**Table 1 ijms-22-06111-t001:** Summary of known Semaphorin (SEMA) and Plexin (PLXN) contributions to neurodegenerative disease pathophysiology and development. Note that the focus here is brought to dysregulations found in the brain. Other Semaphorins and Plexins in immune cells are found and likely contribute to neuroinflammatory characteristics of these diseases.

Disease	SEMAs/PLXNs	Involvement	Refs
Multiple Sclerosis	SEMA3F	Upregulated, recruits oligodendrocyte precursor cells toward the lesion	[101,103]
	SEMA3A SEMA7A	Upregulated, restrict oligodendrocyte precursor cell recruitment and proliferation	[99,100,101,102,103,140,141]
	PLXNA1/NRP1	Overexpressed in oligodendrocytes	[141,142]
Spinal muscular atrophy	PLXND1	Cleaved in Spinal muscular atrophy	[106]
Parkinson’s disease	SEMA3A	Upregulated in dopaminergic neurons, induces apoptosis	[143]
	SEMA3A/3C	Intranigral expression rescues motor-related deficits and enhances dopamine release	[109]
	SEMA5A	Polymorphism may be a risk gene (used as biomarker). Results are discordant (likely depending on population background)	[108,110,111,144,145]
Huntington disease	SEMA4D	Blockade improves biological, behavioral and cognitive defects	[107]

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
