# Peer review of "Modelling and Refining Neuronal Circuits with Guidance Cues: Involvement of Semaphorins"

_ijms, 2021, doi:10.3390/ijms22116111_

Round 1
Reviewer 1 Report
I found this as a nice review and the models covered are represented well. Others I feel will cite this review as a good source of compiled information on the topic of semaphorins and developmental processes.
There are just a few minor edits to consider:
- Line 57
Drosophila Melanogaster
Change to Drosophila melanogaster (Italic)
- Line 104 "there. interconnected vertically and horizontally" needs to be adjusted.
- At the end
"I apologize with colleague whose works"
Needs to be adjusted. Maybe to "I apologize to colleagues whose works…."
Author Response
I thank the reviewer for the comments.
All of the editing suggested have been done and manuscript proofread for English improvement.
Reviewer 2 Report
This is a well-researched review manuscript on the role of semaphorins in the neural circuit signaling. The only comment that I have is to proofread the manuscript again. There are grammatical errors and several sentences were not grammatically sound. Please see lines 102-104 as one example.
Author Response
I thank the reviewer for constructive comments.
The manuscript has been proofread by native English speakers for improvements.
Reviewer 3 Report
In this review, G. Limoni discusses the involvement of semaphorins during the development of the CNS. Specifically, the author describes the function of semaphorins in neuronal maturation, synaptic plasticity, axon pruning, synaptic homeostasis at neuromuscular junctions, and finally describes their functions in neurodegenerative diseases.
Overall the text is clear and the ideas well organized. However, the English needs to be improved and the following comments need to be addressed to make this review acceptable for publication:
Major comments:
- A short section should include a description of the complex semaphoring-neuropilin-plexin and describe briefly their structure and general function to help the reader understand the rest of the text.
- The font in figure 2 is too small and almost impossible to read. For this figure and for clarity purpose, separate the different sections by a letter (A, B, C etc) and reference each section in the text.
- A figure summarizing the known functions of SEMAs and pathways involved in neurological disorders would help inexperienced readers have a synthetic idea of the current state of the knowledge about SEMAs in these pathologies.
Minor comments:
- Line 51-54: consider revising the sentence by replacing “in the specific” by “specifically.
- Line 79: replace “perturbing” by “disturbing”.
- Line 95: replace “of which expression” by “the expression of which” or “which expression”.
- Line 104: remove the “ .” by a “,” after “there”.
- Line 111: correct “deemonstrated”.
- Line 136: correct “other receptors acts”.
- Line 145: correct “formed stabilized”.
- Line 156: correct “weeree” and replace “contribute in” by “contribute to”.
- Line 168: replace “implied” by “involved”.
- Line 177: replace “shown as involved” by shown to be involved” and “in the specific” by “specifically”.
- Line 188: correct “this reveal”.
- Line 191: correct “SEMA3B and SEMA3F has been”.
- Line 206: remove the “.” After “Although the”.
- Line 238: correct “the mechanisms an”.
- Line 240: replaced “enriched of” by “enriched in”.
- Line 355-357: this sentence is vague and needs to be written correctly.
Author Response
I thank the reviewer for the constructive comments.
The manuscript has been revised and English proofread and improved.
I have added a "Box" explaining briefly Semaphorins and Plexins interactions, as asked.
Fonts in Figure 2 have been corrected and letters for different panels added as suggested.
Finally, a Table summarizing Sempahorins and Plexins known to be involved in neurodegenerative has been added as overview.
Reviewer 4 Report
Present paper deals on the role of semaphorins during the brain development and in the maintenance of synaptic connectivity through adult life. In my opinion the paper is concise, well written, and shows the reader an overall situation on the art state about semaphorins and neurodegenerative (ALS) diseases. Therefore, it is suitable for publication in present form.
Author Response
I thank the reviewer for the comments.
The manuscript has additionally been proofread for English improvement and other changes included in the final version.